# Single-Port Robotic Posterior Retroperitoneoscopic Adrenalectomy: Current Perspectives, Technical Considerations, and Future Directions

**DOI:** 10.3390/jcm14072314

**Published:** 2025-03-28

**Authors:** Kwangsoon Kim

**Affiliations:** Department of Surgery, College of Medicine, The Catholic University of Korea, Seoul 06591, Republic of Korea; noar99@naver.com

**Keywords:** robotic surgery, single-port robotic system, retroperitoneal adrenalectomy, minimally invasive surgery

## Abstract

Single-port (SP) robotic posterior retroperitoneoscopic adrenalectomy (SP-PRA) represents a State-of-the-Art innovation in endocrine surgery, offering a minimally invasive approach for adrenal gland resection with significant improvements in surgical precision, cosmetic outcomes, and patient quality of life. The SP robotic system facilitates surgery through a single incision in the back, avoiding the transperitoneal cavity and enabling direct retroperitoneal access to the adrenal gland. This review explores the evolution, techniques, and clinical outcomes of SP-PRA, emphasizing its advantages over traditional multi-port and laparoscopic methods. Enhanced visualization and precise articulation of the SP robotic system minimize trauma to surrounding tissues, leading to fewer complications and faster recovery times. Initial studies suggest superior patient satisfaction due to hidden incisions and excellent postoperative outcomes. However, challenges such as a steep learning curve, high costs, and limited long-term data remain. This review highlights the need for continued research and innovation to optimize the adoption of SP-PRA and expand its indications.

## 1. Introduction

Adrenalectomy, the surgical removal of the adrenal glands, has a long and transformative history within the field of surgery [1,2,3,4,5]. First described in the late 19th century, open adrenalectomy was the initial approach for treating adrenal tumors and hyperfunctional conditions [4,6,7,8,9,10]. The procedure, although effective in tumor removal, was highly invasive, involving large incisions and significant disruption to surrounding tissues [3,11,12]. The associated challenges included considerable postoperative pain, prolonged recovery times, and visible scarring, which often discouraged surgical intervention except in life-threatening cases.

The advent of laparoscopic adrenalectomy in the early 1990s marked a pivotal advancement in adrenal surgery [13]. This minimally invasive technique significantly reduced patient morbidity by decreasing postoperative pain, shortening hospital stays, accelerating recovery, and enhancing cosmetic outcomes. The transperitoneal laparoscopic approach, in particular, provided surgeons with superior visualization of the abdominal cavity, rendering it especially effective for resecting smaller tumors or those situated in the upper regions of the adrenal gland [14,15,16,17]. Despite these benefits, laparoscopic adrenalectomy posed certain challenges. The limited maneuverability of instruments inherent to laparoscopy and the requisite two-dimensional visualization demanded a steep learning curve for surgeons. Moreover, the procedure’s complexity increased with larger or more intricate tumors, often necessitating conversion to open surgery, which could diminish the minimally invasive advantages of the laparoscopic approach [18,19,20,21].

In response to the challenges associated with laparoscopic adrenalectomy, posterior retroperitoneoscopic adrenalectomy (PRA) was introduced as an alternative technique. This approach provides direct access to the adrenal glands through the retroperitoneal space, thereby eliminating the need to traverse the peritoneal cavity [22,23,24,25,26,27,28]. This technique significantly mitigates the risk of intra-abdominal adhesions and expedites postoperative recovery. It is particularly advantageous for patients with a history of abdominal surgeries, where existing adhesions pose substantial risks [25,26,27,29,30,31]. However, the utility of PRA is constrained by the limited working space inherent to the retroperitoneal approach and the technical challenges associated with navigating complex retroperitoneal anatomy using conventional laparoscopic instruments. These factors necessitate a high degree of surgical expertise and can impede the widespread adoption of PRA [24,27,32,33].

The advent of robotic-assisted surgery in the early 2000s revolutionized adrenalectomy, effectively addressing many limitations inherent in both laparoscopic and open approaches [34,35,36,37,38]. The multi-port robotic systems, such as the da Vinci Surgical System, provided three-dimensional high-definition (3D HD) visualization, enhanced dexterity through articulated instruments, and tremor filtration. These advancements facilitated precise dissection and reduced complications in adrenal surgery [36,39,40]. Nevertheless, multi-port robotic adrenalectomy necessitates multiple incisions, which can increase tissue trauma and potentially diminish some of the cosmetic benefits associated with minimally invasive surgery.

The advent of single-port (SP) robotic technology signifies a significant advancement in adrenal surgery. Unlike traditional multi-port systems, the SP robotic platform integrates all surgical instruments and the camera into a single cannula. This innovation enables surgeons to perform complex procedures through a single small incision, minimizing tissue disruption and optimizing cosmetic outcomes [41,42,43,44]. The SP robotic system is particularly advantageous for retroperitoneoscopic adrenalectomy, where the confined anatomical space benefits from the system’s advanced visualization and precise instrumentation. This approach significantly reduces postoperative pain and scarring, thereby enhancing patient satisfaction [40,44,45].

The transition to SP robotic posterior retroperitoneoscopic adrenalectomy (SP-PRA) represents a significant advancement in adrenal surgery, effectively combining the minimally invasive benefits of retroperitoneoscopic access with the technological innovations of the SP robotic system. This approach enhances surgical precision and patient outcomes by minimizing tissue disruption and optimizing cosmetic results [40,45,46,47,48]. The single incision, strategically placed in the back, provides direct access to the adrenal glands without breaching the peritoneal cavity. Utilizing the SP robotic system minimizes tissue trauma, while the platform’s enhanced dexterity and superior visualization capabilities facilitate precise tumor resection and meticulous hemostasis.

SP-PRA offers several advantages in terms of patient recovery and outcomes. The utilization of a single incision minimizes postoperative pain and reduces the risk of complications such as hernias or infections. This approach has been associated with shorter hospital stays and expedited return to normal activities. Additionally, the concealed location of the incision enhances cosmetic outcomes, which is particularly beneficial for patients concerned about visible scarring [40,45,46].

Despite its numerous advantages, SP-PRA presents certain challenges. The adoption of this technique is impeded by a steep learning curve, as surgeons must adapt to operating within a confined space using specialized SP instrumentation [40,45,49]. The high cost of the robotic platform and associated equipment also poses barriers to widespread adoption, particularly in resource-limited settings. Furthermore, long-term data on oncological and functional outcomes are limited, necessitating further research to validate the technique’s efficacy and safety in diverse patient populations.

The evolution of adrenalectomy underscores an enduring commitment to advancing surgical techniques, prioritizing enhanced patient outcomes while minimizing procedural invasiveness. From the foundational practices of open adrenal surgery to the transformative innovations of laparoscopic, retroperitoneoscopic, and robotic methodologies, each progression has played a pivotal role in refining adrenal surgical care. The development of SP-PRA represents the culmination of these efforts, offering a State-of-the-Art solution that harmonizes surgical precision, superior cosmetic results, and heightened patient satisfaction.

The primary objective of this review is to provide a comprehensive evaluation of SP-PRA, including its technical advancements, surgical feasibility, perioperative outcomes, and potential limitations. By analyzing the existing literature and clinical data, we aim to assess the effectiveness of SP-PRA in improving patient outcomes, reducing surgical morbidity, and expanding the indications for robotic adrenalectomy. Furthermore, we highlight current gaps in knowledge and propose future research directions to enhance the widespread adoption and optimization of this technique.

## 2. Types of Robotic Adrenalectomy

Each type of robotic adrenalectomy has unique advantages and limitations. Surgeons must consider patient-specific factors, including tumor size, location, and prior surgical history, to select the most appropriate approach. Each type of robotic adrenalectomy is summarized in Table 1.

Robotic adrenalectomy has undergone significant advancements, leading to the development of various approaches tailored to specific patient and disease characteristics. These approaches can be categorized into four main types, each with distinct technical and clinical implications.

The transperitoneal multi-port approach was one of the earliest applications of robotic technology in adrenal surgery. This method involves accessing the adrenal gland through the peritoneal cavity with multiple ports. The transperitoneal route offers wide working space and excellent visualization of the surgical field, making it ideal for large or complex adrenal tumors [18,36,50,51,52,53]. However, the requirement for multiple ports increases tissue trauma, leading to more postoperative pain and longer recovery periods.

In contrast, the posterior retroperitoneal multi-port approach provides direct access to the adrenal glands without traversing the peritoneal cavity. By avoiding the peritoneum, this method reduces the risk of intra-abdominal adhesions and accelerates recovery. The posterior retroperitoneal route is particularly advantageous for patients with prior abdominal surgeries, where adhesions may complicate a transperitoneal approach [54,55,56,57,58,59]. However, the confined working space and reliance on multiple ports can pose challenges, especially for surgeons unfamiliar with retroperitoneal anatomy.

The introduction of single-port technology has further refined robotic adrenalectomy. The SP transperitoneal approach combines the minimally invasive benefits of single-port access with the extensive visualization capabilities of the transperitoneal route. By reducing the number of incisions to one, this method minimizes tissue disruption and improves cosmetic outcomes [44,60]. Nevertheless, instrument crowding within the single port and limited long-term outcome data remain challenges that warrant further research.

The most recent approach is the SP-PRA. This technique involves creating a single incision in the back to access the retroperitoneal space, offering unparalleled cosmetic outcomes and reduced postoperative pain. The SP robotic system’s articulated instruments and high-definition visualization facilitate precise dissection and tumor resection, even in anatomically challenging cases. While SP-PRA shows promise in improving recovery times and patient satisfaction, it is associated with a steep learning curve and limited data on long-term oncological outcomes [40,44,45,47]. Expanding the indications for SP-PRA to include larger and more complex tumors will require additional studies and technical refinements.

In summary, the evolution of robotic adrenalectomy highlights the continuous pursuit of minimizing invasiveness while optimizing surgical outcomes. Each approach offers unique advantages and limitations, necessitating careful patient selection based on tumor size, location, and surgeon expertise. Future advancements in robotic technology and clinical research will likely expand the applicability and improve the efficacy of these techniques, solidifying their role in modern endocrine surgery.

## 3. Overview of the SP Robotic System

The SP robotic system is a groundbreaking advancement in robotic-assisted surgery, specifically designed for single-port access procedures. Unlike traditional multi-port systems that require multiple ports, the SP system operates through a single cannula, which houses a 3D HD camera and three multi-jointed surgical instruments. This innovative design minimizes the number and size of incisions, substantially reducing tissue trauma and enhancing postoperative recovery [61,62,63,64,65,66].

The single-port design of the system allows for unparalleled precision in confined spaces, making it particularly suitable for retroperitoneal surgeries such as adrenalectomy. The fully wristed instruments mimic the dexterity of the human hand, enabling complex surgical maneuvers that were previously challenging in minimally invasive settings. Enhanced visualization provided by the system’s 3D HD camera ensures accurate identification and preservation of vital structures, such as adrenal veins and surrounding vasculature.

This system’s ergonomic console provides intuitive controls and real-time feedback, allowing surgeons to perform procedures with greater ease and less fatigue. Its application is not limited to adrenal surgery; the SP system is also utilized in other specialties, including urology and general surgery, for procedures like prostatectomy and retroperitoneal lymphadenectomy [61,62,67,68,69,70,71]. Its versatility and precision make it a transformative tool in the field of minimally invasive surgery, contributing to improved outcomes and patient satisfaction.

## 4. Patient Selection Criteria

Proper patient selection is critical for the success of SP-PRA. Ideal candidates include patients with small to medium-sized adrenal tumors (<5 cm) without extensive vascular involvement or surrounding organ adhesion. Functional adrenal tumors, such as pheochromocytoma and aldosterone-producing adenomas, and non-functional adenomas are particularly well-suited for this technique [11,72,73]. Preoperative imaging, including CT or MRI, plays a vital role in assessing tumor size, location, and resectability.

Patients with prior retroperitoneal surgery or significant comorbidities require careful evaluation, and alternative approaches may be considered in cases involving large tumors (>10 cm) or complex vascular anatomy. As surgical teams gain experience with SP-PRA, the criteria for patient selection are expected to broaden, potentially encompassing larger and more challenging tumors. Enhanced understanding of anatomical variations and advancements in robotic technology will likely facilitate these expanded indications.

## 5. Surgical Procedures

The surgical technique for SP-PRA is highly refined and follows a sequence designed to optimize safety, efficiency, and patient outcomes. Detailed surgical techniques have been described in previous studies, but this section expands on the critical steps that contribute to the success of this procedure [40,45,54,74].

The procedure begins with patient positioning. Under general anesthesia, the patient is placed in the prone position, lying on a soft support bar positioned below both the anterior and superior iliac spines. This configuration ensures optimal exposure to the retroperitoneal space while minimizing the risk of pressure injuries. The operating table is tilted approximately 10 degrees head-down to facilitate the gravitational displacement of intra-abdominal contents away from the surgical site.

A 3–4 cm skin incision is made just below the lowest tip of the 11th and 12th ribs (Figure 1). This incision serves as the entry point for the SP robotic cannula. Through this single incision, subcutaneous and muscle layers are dissected meticulously to create a retroperitoneal working space. Maintaining minimal tissue trauma during this step is critical to preserving the integrity of surrounding structures and reducing postoperative pain. To maintain the working space, carbon dioxide (CO_2_) gas is insufflated at a pressure of 18 mmHg, ensuring adequate visualization and maneuverability.

Docking the SP robotic system is the next critical step. The SP cannula, which houses a 3D HD camera and three multi-jointed robotic instruments, is inserted into the incision (Figure 2). The camera offers enhanced visualization, providing surgeons an immersive and detailed view of the retroperitoneal anatomy. The robotic instruments, controlled intuitively from the surgeon’s console, are designed to mimic the natural dexterity of the human hand. Their multi-jointed articulation allows for precise dissection and manipulation in the confined retroperitoneal space.

The dissection phase involves meticulous separation of the adrenal gland from its surrounding tissues. The connective tissues anchoring the gland are carefully divided, with particular attention paid to preserving critical structures, such as the inferior vena cava, kidney, and adjacent organs. The adrenal veins are ligated securely using advanced robotic tools or clips, ensuring hemostasis while minimizing the risk of vascular injury. The HD visualization provided by the robotic system is particularly advantageous during this phase, as it enhances the surgeon’s ability to navigate complex anatomical variations.

Once the adrenal gland is mobilized, it is extracted intact through a single incision using an endo-pouch. Ensuring the gland’s integrity during extraction is vital for accurate histopathological analysis, especially in cases of suspected malignancy. The surgical site is then irrigated thoroughly to remove any residual debris or blood, and meticulous hemostasis is achieved to minimize the risk of postoperative complications.

Closure of the surgical site marks the final stage of the procedure. The single incision is closed using sutures or surgical adhesive, with care taken to achieve optimal cosmetic results. This streamlined approach not only enhances the aesthetic outcome but also contributes to faster healing and reduced risk of wound-related complications.

SP-PRA exemplifies the integration of advanced robotic technology with minimally invasive surgical principles. Each step of the procedure is designed to maximize safety, efficiency, and patient satisfaction, making it a cornerstone technique in modern adrenal surgery.

## 6. Clinical Outcomes

### 6.1. Surgical Outcomes

SP-PRA has demonstrated surgical outcomes comparable to traditional laparoscopic and multi-port robotic techniques. The complete resection of adrenal tumors with negative surgical margins is a critical objective in adrenal surgery, particularly for functional tumors such as pheochromocytomas and cortisol-producing adenomas. The enhanced 3D visualization provided by the SP robotic system facilitates meticulous dissection and precise margin control. Studies have shown that SP-PRA achieves oncological clearance rates on par with multi-port robotic systems, with no increase in tumor recurrence rates over short- to mid-term follow-up periods [40,45]. This level of oncological safety extends to complex cases involving adrenal cortical carcinoma or metastatic adrenal lesions. Although SP-PRA is typically reserved for small to medium-sized tumors (<5 cm), as experience and technical expertise grow, indications may expand to include larger and more invasive tumors. The ability of the SP robotic system to navigate narrow retroperitoneal spaces without compromising surgical field visibility makes it a promising tool for these challenging cases.

### 6.2. Complications

The SP-PRA procedure is associated with a low incidence of complications, including intraoperative bleeding, organ injury, and postoperative infections. The precise control and improved visualization offered by the SP robotic system reduce the likelihood of vascular and tissue trauma. Adrenal veins, which are critical to the procedure, can be safely ligated with minimal risk of hemorrhage, even in hypervascular tumors such as pheochromocytomas.

Postoperative complications, including hematoma, wound infection, and delayed healing, are rare. Comparative studies have indicated that SP-PRA has a lower complication rate than open adrenalectomy and is comparable to other minimally invasive techniques [25,33]. Additionally, the SP-PRA minimizes the risk of hernias, a complication sometimes associated with multi-port surgeries.

### 6.3. Cosmetic Outcomes

Cosmetic outcomes are a defining strength of SP-PRA. The use of a single incision hidden within the natural contour of the back enhances cosmetic results. This feature is particularly significant for younger patients or individuals for whom visible scarring might impact quality of life. Patient satisfaction surveys consistently rank SP robotic surgery highly in terms of aesthetic outcomes, surpassing both laparoscopic and multi-port robotic techniques [75,76,77].

The psychological benefits of improved cosmetic outcomes should not be underestimated. Reduced visible scarring can enhance patients’ overall satisfaction with their surgical experience and contribute to positive mental health outcomes during recovery.

### 6.4. Recovery and Quality of Life

SP-PRA is associated with faster recovery times compared to traditional approaches. Patients benefit from reduced postoperative pain, shorter hospital stays, and quicker returns to normal activities [42,61,62,78]. The minimally invasive nature of the procedure reduces physical strain on the body, facilitating an accelerated healing process. Pain management is another area where SP-PRA shows significant advantages. The smaller incision size results in reduced postoperative discomfort, often requiring less opioid analgesia. This reduction in pain medication usage not only lowers the risk of opioid dependence but also minimizes side effects such as nausea and sedation, which can delay recovery.

### 6.5. Comparative Studies

Comparative studies between SP-PRA, multi-port robotic adrenalectomy, and laparoscopic adrenalectomy provide critical insights into the procedure’s efficacy and outcomes. In head-to-head comparisons, SP-PRA consistently matches other techniques in terms of operative time, blood loss, complication rates, and recovery metrics [40,45].

Longitudinal studies tracking outcomes over five or more years will be essential to validating SP-PRA’s benefits in terms of tumor recurrence, postoperative pain, and overall outcomes. These data will be critical as the procedure is adopted more widely and applied to increasingly complex cases.

### 6.6. Future Directions

As experience with SP-PRA grows, its applications are expected to expand. Current contraindications, such as tumor size >5 cm or significant vascular involvement, may become relative rather than absolute barriers. Advanced robotic technology, such as augmented reality-assisted navigation and improved imaging modalities, could enhance the feasibility of SP-PRA for larger and more invasive tumors. Additionally, the refinement of training programs for surgeons will play a pivotal role in optimizing outcomes. The steep learning curve associated with SP-PRA underscores the importance of structured proctoring and simulation-based training. As more surgeons become proficient in the technique, procedural times and complication rates are likely to decrease further.

The role of patient-reported outcomes in future studies will also be critical. Incorporating metrics such as pain scores, functional recovery timelines, and quality-of-life indices into clinical trials will provide a more comprehensive understanding of the patient experience.

## 7. Discussion

SP-PRA represents a significant advancement in the field of adrenal surgery, providing a minimally invasive alternative that effectively balances cosmetic benefits with oncological and functional efficacy. This innovative approach leverages cutting-edge robotic technology to overcome many limitations of traditional multi-port and laparoscopic techniques while offering superior patient outcomes.

One of the most striking advantages of SP-PRA is its ability to achieve surgical outcomes comparable to multi-port and laparoscopic techniques while significantly improving cosmetic results. Complete tumor resection with negative surgical margins remains the cornerstone of adrenal surgery. The advanced 3D visualization and precise articulation of robotic instruments in SP-PRA enable meticulous dissection, even in anatomically complex retroperitoneal spaces, enhancing surgical accuracy and safety. Studies have demonstrated that SP-PRA does not compromise surgical effectiveness for small to medium-sized adrenal tumors treated with this approach. These findings reinforce its viability as a safe and effective surgical option, offering patients faster recovery, less postoperative pain, and superior aesthetic outcomes compared to traditional techniques [40,45].

The feasibility of SP-PRA in patients with a history of prior abdominal surgery or retroperitoneal fibrosis remains an important consideration, as these conditions may present anatomical challenges that impact retroperitoneal access. In patients with previous abdominal operations, extensive intra-abdominal adhesions can complicate transperitoneal approaches, making retroperitoneoscopic access a preferred alternative. SP-PRA offers the advantage of avoiding the peritoneal cavity, reducing the risk of adhesiolysis-related complications, and improving postoperative recovery. However, in cases of significant retroperitoneal scarring or fibrosis, often encountered in patients with a history of chronic inflammation, radiation therapy, or repeated surgical interventions, dissection within the confined retroperitoneal space may be technically demanding. In such cases, preoperative imaging is essential for evaluating the extent of fibrosis and planning surgical strategies accordingly. Surgeons performing SP-PRA in these patients must have a high level of expertise in retroperitoneoscopic dissection, and careful patient selection is crucial to ensure optimal outcomes.

Cosmetic outcomes play a crucial role in patient satisfaction, particularly for younger or cosmetically sensitive individuals. The SP approach reduces visible scarring, offering near-scarless results by concealing the incision in the natural folds of the back. This feature not only improves the physical appearance but also has psychological benefits. Patients often experience improved self-esteem and confidence, contributing to an overall positive postoperative experience and enhanced quality of life during recovery.

Beyond aesthetics, SP-PRA significantly improves functional outcomes by reducing recovery times and postoperative discomfort compared to conventional surgical methods. The minimally invasive nature of SP-PRA, characterized by smaller incisions and less tissue disruption, accelerates healing and minimizes discomfort. Studies have shown that many individuals report a faster return to work and normal routines, underscoring the procedure’s efficiency in promoting recovery [79,80]. Furthermore, the reduced surgical trauma associated with SP-PRA minimizes the need for opioid analgesics, thereby lowering the risk of dependency and adverse side effects, such as nausea and constipation. This benefit further enhances patient safety and long-term health outcomes.

Despite the numerous advantages of SP-PRA, its widespread adoption is hindered by several challenges, primarily the steep learning curve associated with the SP robotic system. Operating within the confined retroperitoneal space requires a high level of technical skill, as surgeons must navigate complex anatomical structures while managing instrument triangulation and optimizing visualization through a single access point. The limited workspace increases the complexity of performing delicate maneuvers, necessitating enhanced hand–eye coordination and precise instrument control. To address these challenges, structured training programs, mentorship opportunities, and simulation-based educational platforms are essential to overcome these hurdles. These resources provide surgeons with the necessary experience to build confidence and proficiency, ensuring consistent procedural outcomes across diverse clinical settings. Additionally, the integration of advanced imaging technologies and augmented reality systems may further aid in improving spatial awareness and surgical precision, ultimately contributing to better patient safety and long-term outcomes. As experience with SP-PRA grows and more institutions invest in specialized training initiatives, the adoption of this technique is expected to expand, paving the way for broader acceptance in the field of adrenal surgery. Moreover, its applicability remains limited in certain clinical scenarios. In particular, its role in the management of adrenocortical carcinoma or large adrenal tumors (>6 cm) has not been fully established. These cases often require en bloc resection with clear margins and thorough lymphadenectomy, which may be difficult to achieve through SP-PRA. Therefore, open surgery or multi-port robotic adrenalectomy remains the standard of care in such situations. Further studies are necessary to assess the safety, feasibility, and oncologic adequacy of SP-PRA in these challenging cases.

Cost remains a significant barrier to the broader adoption of SP-PRA, particularly in smaller hospitals and resource-constrained healthcare systems. The substantial upfront investment required for acquiring robotic platforms, combined with ongoing maintenance expenses and the cost of specialized consumables, presents financial challenges that may deter institutions from adopting this advanced technology. Furthermore, the need for dedicated surgical teams trained in robotic techniques adds to the overall expense, further limiting accessibility. Cost-effectiveness analyses are crucial to fully assess the financial viability of SP-PRA in comparison to multi-port robotic and laparoscopic adrenalectomy. These studies should consider not only procedural costs but also long-term benefits such as reduced complication rates, shorter hospital stays, and improved quality of life to provide a comprehensive understanding of the economic impact of SP-PRA. Addressing these financial challenges through strategic investment, potential reimbursement policies, and broader adoption of robotic technology may help facilitate more widespread implementation of SP-PRA in the future.

As the adoption of SP-PRA continues to expand, the criteria for patient selection are expected to evolve, accommodating a broader range of clinical scenarios. Currently, the procedure is primarily indicated for patients with small to medium-sized adrenal tumors that lack significant vascular involvement or local invasion. These criteria help ensure optimal surgical outcomes and minimize potential complications. However, with continuous advancements in imaging modalities, such as high-resolution ultrasound, multiparametric MRI, and augmented reality-guided navigation, the feasibility of SP-PRA for larger and more complex adrenal tumors is becoming increasingly viable. Improved preoperative planning, combined with enhanced robotic capabilities and refined surgical techniques, may allow for the safe resection of tumors previously deemed unsuitable for this approach. As a result, indications for SP-PRA could extend to include patients with larger lesions, multifocal disease, or tumors invading critical structures. To achieve this goal, rigorous clinical trials and multicenter studies are essential to validate the procedure’s safety, efficacy, and long-term surgical outcomes. Special attention should be given to high-risk patient populations, such as those with recurrent disease, functional adrenal tumors, or significant comorbidities, to establish evidence-based guidelines for the optimal use of SP-PRA in diverse clinical settings.

Long-term outcomes and patient-reported metrics are essential to gaining a comprehensive understanding of the true impact of SP-PRA on patient health and quality of life. While early clinical data suggest favorable short-term outcomes, the long-term outcomes must be thoroughly evaluated through robust, longitudinal studies. These studies will help identify potential late complications, assess disease-free survival rates, and determine the procedure’s effectiveness in a variety of patient populations. Additionally, the integration of patient-reported outcome measures, including pain scores, functional recovery timelines, quality of life assessments, and satisfaction surveys, will provide valuable insights into the patient’s perspective and overall experience. Tracking functional outcomes such as hormonal normalization, return to normal activities, and postoperative discomfort will offer a more comprehensive perspective of SP-PRA’s success beyond traditional clinical metrics. Moreover, comparative studies between SP-PRA and conventional approaches, such as multi-port robotic and laparoscopic adrenalectomy, are crucial to establishing the true value of this technique in routine clinical practice. Collecting and analyzing such comprehensive data will ultimately inform evidence-based guidelines and refine patient selection criteria, optimizing the use of SP-PRA in endocrine surgery.

The integration of artificial intelligence and augmented reality into the SP robotic system holds immense potential for transforming SP-PRA outcomes by enhancing surgical precision, efficiency, and safety. Artificial intelligence can support real-time decision-making, enabling surgeons to identify critical structures with greater accuracy, predict potential complications, and optimize surgical planning. This not only enhances procedural accuracy but also reduces operative times and minimizes the risk of errors. Augmented reality, on the other hand, can provide enhanced visualization by overlaying preoperative imaging data onto the surgical field in real time, offering a more comprehensive understanding of anatomical relationships and aiding in the meticulous dissection of complex structures. These technological advancements could significantly expand the indications for SP-PRA and improve outcomes for patients with challenging adrenal pathologies. However, the successful integration of these technologies into routine clinical practice requires close collaboration between surgeons, researchers, and industry partners. Multidisciplinary efforts are essential to address technical challenges, refine user interfaces, and ensure seamless integration with existing surgical workflows. Additionally, establishing standardized protocols and evidence-based guidelines will be crucial to enhancing the consistency and reproducibility of surgical outcomes across various institutions. Structured training programs and educational initiatives are essential to equip surgeons with the necessary proficiency in these emerging technologies, ensuring their effective implementation in clinical practice.

The future directions for SP-PRA encompass a range of innovative advancements aimed at optimizing surgical precision, improving patient outcomes, and expanding its clinical applications. The incorporation of machine learning algorithms holds great potential for predicting patient-specific risks, refining surgical planning, and enabling real-time intraoperative decision-making, ultimately enhancing procedural safety and efficiency. Additionally, personalized medicine approaches, facilitated by the integration of genomic, imaging, and biochemical data, could allow for tailored surgical interventions based on individual patient profiles, leading to improved treatment outcomes and reduced postoperative complications. Further research into enhancing the ergonomics of robotic systems will be critical to improving surgeon comfort and efficiency during lengthy procedures, reducing fatigue, and enhancing precision. Expanding the compatibility of the SP robotic system with adjunctive technologies such as intraoperative ultrasound, fluorescence imaging, and augmented reality-guided navigation will provide surgeons with superior visualization capabilities and facilitate more accurate dissection of critical structures. These advancements will not only reinforce SP-PRA’s role as a transformative surgical technique but also pave the way for broader adoption in endocrine surgery and other minimally invasive procedures.

Despite these promising perspectives, current evidence on the long-term oncologic efficacy of SP-PRA remains limited. There is a lack of prospective or randomized studies evaluating recurrence rates, disease-free survival, or overall survival following SP-PRA. As such, the oncologic safety of this technique, particularly in malignant adrenal tumors, requires further validation through long-term follow-up and multicenter research.

## 8. Conclusions

SP-PRA is a groundbreaking advancement in endocrine surgery, combining minimally invasive techniques with enhanced cosmetic and functional outcomes. While SP-PRA represents an important advancement in adrenal surgery, further prospective, multicenter studies are necessary to establish its long-term oncologic outcomes, cost-effectiveness, and optimal patient selection criteria. As research continues to evolve, SP-PRA may become a preferred technique for selected patients, but additional data are needed before it can be widely considered a standard of care.

## Figures and Tables

**Figure 1 jcm-14-02314-f001:**
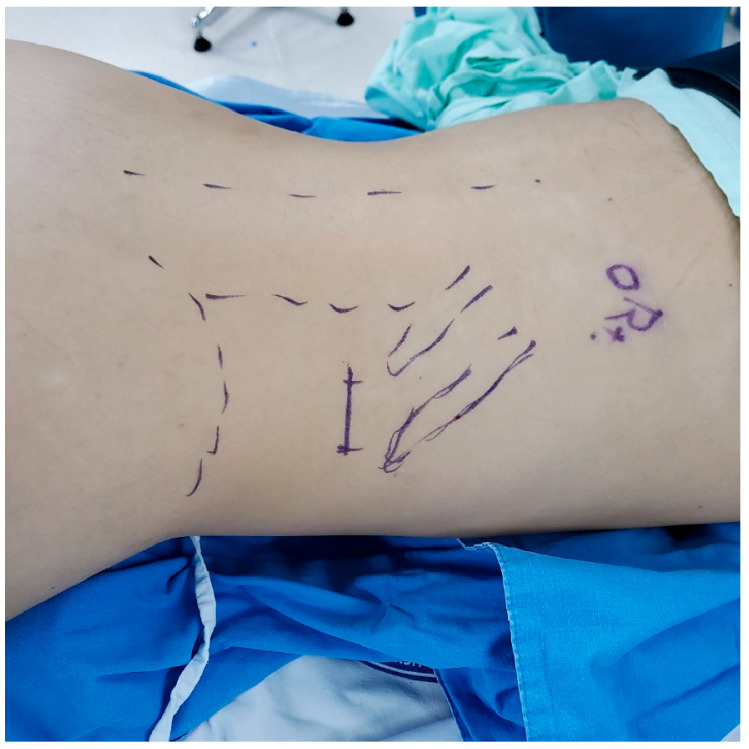
Skin incision below the lowest tip of the 11th and 12th ribs.

**Figure 2 jcm-14-02314-f002:**
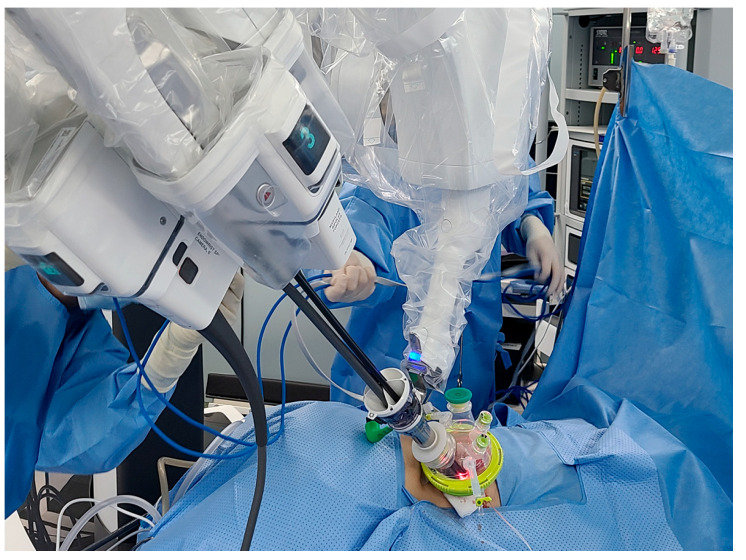
Docking with 3D HD camera and 3 robotic instruments.

**Table 1 jcm-14-02314-t001:** Types of robotic adrenalectomy.

Type	Approach Description	Advantages	Limitations
Transperitoneal Multi-Port	Accesses the adrenal gland through the peritoneal cavity with multiple ports.	1. Wide working space; 2. Excellent visualization.	1. Multiple incisions; 2. Increased tissue trauma.
Posterior Retroperitoneal Multi-Port	Direct retroperitoneal access with multiple ports.	1. Avoids peritoneal cavity; 2. Reduced risk of adhesions.	1. Requires multiple incisions; 2. Steep learning curve for confined spaces.
Single-Port Transperitoneal	Combines transperitoneal approach with single-port technology.	1. Minimally invasive; 2. Reduces incision-related morbidity.	1. Limited by instrument crowding; 2. Limited long-term outcome data.
Single-Port Posterior Retroperitoneoscopic (SP-PRA)	Latest innovation with single back incision for retroperitoneal access.	1. Avoids peritoneal cavity; 2. Minimally invasive; 3. Best cosmetic outcomes; 4. Reduced pain; 5. Faster recovery.	1. Steep learning curve; 2. Limited long-term outcome data; 3. Challenging for large tumors;

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
