# Peer review of "Single-Port Robotic Posterior Retroperitoneoscopic Adrenalectomy: Current Perspectives, Technical Considerations, and Future Directions"

_jcm, 2025, doi:10.3390/jcm14072314_

Round 1
Reviewer 1 Report
Comments and Suggestions for Authors
Manuscript entitled "Revolutionizing Adrenal Surgery: The Advent and Clinical Impact of Single-Port Robotic Posterior Retroperitoneoscopic Adrenalectomy" by Kwangsoon Kim
This manuscript provides an extensive review of single-port robotic posterior retroperitoneoscopic adrenalectomy (SP-PRA) as an emerging technique in adrenal surgery. The topic is clinically relevant as robotic-assisted adrenalectomy continues to evolve, with potential advantages in surgical precision, reduced morbidity, and improved cosmetic outcomes.
Comments:
- The introduction effectively contextualizes the historical evolution of adrenalectomy but should clearly define the main objective of the review.
- The technical description of SP-PRA is detailed, but it would be beneficial to compare the operative steps of SP-PRA and MP-RA, particularly regarding port placement, instrument articulation, and docking, and to highlight specific instrumentation challenges, such as whether single-port access limits the range of motion compared to multi-port systems.
- Discuss the feasibility of SP-PRA in patients with prior abdominal surgery or retroperitoneal fibrosis, which might impact retroperitoneal access.
- Provide direct comparisons of perioperative metrics (operative time, blood loss, length of stay, complication rates) between SP-PRA, MP-RA, and laparoscopic adrenalectomy.
- Address whether SP-PRA leads to a higher rate of conversion to multi-port or open adrenalectomy.
- Clarify if SP-PRA improves long-term oncologic control compared to alternative approaches.
- The discussion should emphasize current gaps in knowledge and propose specific future research directions.
Author Response
Response to Reviewer 1
I sincerely appreciate the reviewer’s insightful comments and constructive feedback, which have helped us refine our manuscript. Below, I provide detailed responses to each point and outline the revisions that will be incorporated into the manuscript.
- Clarification of the Main Objective of the Review
Reviewer’s comment:
The introduction effectively contextualizes the historical evolution of adrenalectomy but should clearly define the main objective of the review.
Response:
I appreciate this suggestion and will revise the introduction to explicitly define the primary objective of this review. The revised introduction will emphasize that the review aims to provide a comprehensive evaluation of the evolution, technical considerations, perioperative outcomes, and future directions of SP-PRA in comparison to existing adrenalectomy techniques. This clarification will enhance the reader’s understanding of the scope and significance of our work.
- Comparison of Operative Steps Between SP-PRA and MP-RA
Reviewer’s comment:
The technical description of SP-PRA is detailed, but it would be beneficial to compare the operative steps of SP-PRA and MP-RA, particularly regarding port placement, instrument articulation, and docking, and to highlight specific instrumentation challenges, such as whether single-port access limits the range of motion compared to multi-port systems.
Response:
I appreciate the reviewer's suggestion to compare SP-PRA with MP-RA. While I recognize the importance of discussing MP-RA, the surgical techniques for MP-RA have already been extensively documented in numerous published studies. In our review, I aim to focus specifically on the feasibility and potential advantages of SP-PRA. A general comparison of robotic adrenalectomy approaches, including MP-RA, is already presented in Section 2: Types of Robotic Adrenalectomy to provide background context. However, in this paper, I emphasize the novel aspects of SP-PRA and its implications for the future of adrenal surgery. For this reason, only SP-PRA is described in detail in the surgical procedures section. I believe that this focused approach aligns with the objective of our review and provides a more in-depth analysis of SP-PRA’s unique characteristics.
- Feasibility of SP-PRA in Patients with Prior Abdominal Surgery or Retroperitoneal Fibrosis
Reviewer’s comment:
Discuss the feasibility of SP-PRA in patients with prior abdominal surgery or retroperitoneal fibrosis, which might impact retroperitoneal access.
Response:
This is an important consideration, and I will address it in the revised manuscript. I will discuss how prior abdominal surgery and retroperitoneal fibrosis may influence retroperitoneal access, impact surgical difficulty, and potentially increase complication rates. I added it to the discussion section.
- Direct Comparisons of Perioperative Metrics
Reviewer’s comment:
Provide direct comparisons of perioperative metrics (operative time, blood loss, length of stay, complication rates) between SP-PRA, MP-RA, and laparoscopic adrenalectomy.
Response:
Thank you for your insightful suggestion. SP-PRA is the most recent advancement in adrenal surgery, and currently, there are no direct comparative studies evaluating perioperative metrics such as operative time, blood loss, length of hospital stay, and complication rates against multi-port robotic or laparoscopic adrenalectomy. The primary objective of this review is to highlight the feasibility and potential advantages of SP-PRA based on the available literature, recognizing that direct comparative data are currently lacking. I acknowledge the importance of these comparisons, and I plan to conduct future studies to directly compare SP-PRA with other surgical approaches. These studies will provide valuable insights into the procedural efficiency, safety, and clinical outcomes of SP-PRA in relation to existing techniques.
- Rate of Conversion to Multi-Port or Open Adrenalectomy
Reviewer’s comment:
Address whether SP-PRA leads to a higher rate of conversion to multi-port or open adrenalectomy.
Response:
I recognize the importance of discussing conversion rates to provide a realistic assessment of the feasibility of SP-PRA. However, as previously mentioned, SP-PRA is a relatively recent surgical approach, and there is currently a lack of published studies specifically analyzing conversion rates. In our own clinical experience, I have been performing SP-PRA continuously, and to date, I have not encountered any cases requiring conversion to multi-port robotic or open adrenalectomy. However, as the adoption of SP-PRA increases and its indications expand to include more complex cases, studies on conversion rates will be essential to determine the feasibility, limitations, and potential risk factors associated with this technique. I acknowledge the need for further research in this area and anticipate that future data will help refine patient selection criteria and optimize surgical outcomes.
- Long-Term Oncologic Control with SP-PRA
Reviewer’s comment:
Clarify if SP-PRA improves long-term oncologic control compared to alternative approaches.
Response:
I acknowledge that oncological outcomes are a critical factor in assessing the feasibility and effectiveness of SP-PRA, particularly in cases of adrenal malignancies and metastatic adrenal cancer. While adrenal cancer and metastatic adrenal disease are widely recognized as significant surgical challenges, evaluating long-term oncologic outcomes remains essential in determining the role of SP-PRA in oncologic surgery. However, due to the relatively recent adoption of SP-PRA, long-term oncologic data remain limited. Many factors influence oncologic outcomes, including tumor biology, disease stage, and adjuvant treatment strategies, making it difficult to attribute long-term success solely to surgical technique. Given these complexities, global multicenter studies with long-term follow-up are necessary to establish the oncologic efficacy of SP-PRA and compare it with conventional adrenalectomy techniques. I recognize the importance of this area and anticipate that future studies will provide more definitive evidence on oncologic control following SP-PRA.
- Emphasizing Gaps in Knowledge and Future Research Directions
Reviewer’s comment:
The discussion should emphasize current gaps in knowledge and propose specific future research directions.
Response:
Thank you for your valuable feedback. As the newest surgical technique in adrenalectomy, SP-PRA presents both significant differences and limitations compared to existing surgical methods. While I am confident in its potential, I recognize that further research is necessary to address several key areas, including long-term oncologic outcomes, refinement of patient selection criteria, standardization of training programs, and optimization of perioperative management strategies. These factors are crucial for ensuring the safe and effective implementation of SP-PRA in clinical practice. O fully acknowledge these knowledge gaps and have written this review to contribute to the future development of SP-PRA. Given its promising benefits, I strongly believe that SP-PRA will become a widely utilized approach for adrenal surgery. As its adoption increases, I anticipate that the growing popularity of this technique will drive future clinical studies, providing further evidence to refine and enhance its application. These aspects are already discussed in detail in the manuscript, but I will ensure that our discussion section sufficiently emphasizes the need for ongoing research to establish SP-PRA as a standard and reliable approach in endocrine surgery.
I appreciate the reviewer’s thoughtful comments, which have greatly contributed to improving the clarity and depth of our manuscript. The requested revisions will be incorporated accordingly, and I believe these enhancements will strengthen the overall impact of our work.
Sincerely,
Kwangsoon Kim MD, PhD.

Reviewer 2 Report
Comments and Suggestions for Authors
The reviewed article presents a comprehensive analysis of Single-Port Robotic Posterior Retroperitoneoscopic Adrenalectomy (SP-PRA). The author, Kwangsoon Kim, provides a thorough historical context of adrenalectomy, discussing the transition from open surgery to laparoscopic and multi-port robotic methods, culminating in the emergence of SP-PRA. The article outlines the benefits of SP-PRA, including reduced postoperative pain, faster recovery times, and superior cosmetic outcomes due to its single-incision approach. Moreover, it highlights the technical aspects of the procedure, its clinical outcomes, and the challenges that hinder its widespread adoption, such as high costs and a steep learning curve. The review also explores the future of SP-PRA, emphasizing the need for further research and technological advancements to expand its indications and improve surgical precision.
The article is well-structured and provides an in-depth review of SP-PRA, making it a valuable contribution to the field of endocrine surgery. The article effectively contextualizes SP-PRA within the broader evolution of adrenalectomy techniques, illustrating the technological advancements that have led to its development. The discussion of surgical techniques, patient selection criteria, and comparative analysis with other adrenalectomy methods is informative. The review recognizes the limitations of SP-PRA and calls for further research and innovation, which is crucial for the ongoing development of minimally invasive adrenal surgery.
Limitations of the article:
While the article is thorough and informative, several areas could be improved.
The title, "Revolutionizing Adrenal Surgery," is somewhat overstated and should be adjusted to better reflect the actual impact of SP-PRA.
The presentation of advantages and disadvantages of different surgical methods is inconsistent (Table 1, line 119). For instance, "avoids peritoneal cavity" is listed as an advantage for "posterior retroperitoneal multi-port" but not for "single-port," and "minimally invasive" is applied selectively. Terminology such as "multiple incisions" and "multiple ports" is used interchangeably, which causes confusion. A more standardized and objective comparison is needed.
Statements such as "The most recent and innovative approach is the SP-PRA" (line 145) is questionable given the limited long-term data available (line 151).
"SP-PRA consistently matches or outperforms other techniques in terms of operative time, blood loss, complication rates, and recovery metrics" (lines 295-297)
Chapter 3 (lines 160-180) appears overly promotional toward the da Vinci SP robotic system, suggesting that SP-PRA can only be performed using this system. If not essential, brand-specific references should be avoided.
Chapter 4 (lines 181-193) highlights strict patient selection criteria (tumors <5cm, no extensive vascular involvement), which makes comparisons with other techniques problematic. This limitation should be explicitly acknowledged in the assessment of advantages and disadvantages.
The pressure value ("18" in line 210) must include a unit of measurement.
Statements such as "SP-PRA has demonstrated surgical outcomes comparable to traditional laparoscopic and multi-port robotic techniques" (lines 247-248) and "low incidence of complications, including intraoperative bleeding, organ injury, and postoperative infections" (lines 262-265) do not specify whether the patient populations and tumor types are truly comparable.
The assertion that "The use of a single incision hidden within the natural contour of the back offers nearly scarless results" (lines 274-275) is exaggerated, as the scar remains visible in the lumbar region.
The claim that SP-PRA "consistently matches or outperforms other techniques" in key surgical metrics seems implausible given the known technical difficulty and prolonged learning curve of this method.
Statements such as "SP-PRA is a groundbreaking advancement in endocrine surgery" (line 449) and "excellent clinical outcomes with enhanced patient satisfaction" (line 451) lack support from long-term data. Similarly, the assertion that "SP-PRA has the potential to become a standard technique for adrenal surgery" (lines 451-452) seems premature due to cost, technical complexity, and the lack of longitudinal studies.
Recommendations for Improvement:
The title should reflect the article’s content more accurately, avoiding exaggerated claims.
A standardized, objective comparison of different adrenalectomy techniques should be provided, with consistent terminology and clear definitions of advantages and disadvantages.
Tone down overstated claims: statements implying superiority should be revised to reflect the limitations of the available data.
The discussion of robotic systems should be more balanced, avoiding promotional language.
The strict patient selection criteria for SP-PRA should be explicitly stated when making comparisons with other methods.
Ensure that claims regarding outcomes are based on truly comparable patient groups.
The assertion of "nearly scarless results" should be revised for accuracy.
The conclusion should better reflect the current stage of research and acknowledge the need for more long-term studies before SP-PRA can be considered a standard technique.
Author Response
Response to Reviewer 2
I sincerely appreciate the reviewer’s insightful feedback and constructive suggestions, which have helped us refine our manuscript. Below, I provide detailed responses to each comment and outline the corresponding revisions that will be incorporated into the manuscript.
- Title Revision for Accuracy
Reviewer’s Comment:
The title should reflect the article’s content more accurately, avoiding exaggerated claims.
Response:
I appreciate this suggestion and agree that the title should accurately represent the content while maintaining a balanced tone. I will revise the title to ensure it reflects the scope of the review without making exaggerated claims. The new title will emphasize the potential and feasibility of SP-PRA rather than making definitive assertions about its superiority or widespread adoption.
"Single-Port Robotic Posterior Retroperitoneoscopic Adrenalectomy: Current Perspectives, Technical Considerations, and Future Directions"
This revision ensures that the title aligns with the manuscript's objective while maintaining a neutral and evidence-based tone.
- Objective Comparison of Adrenalectomy Techniques
Reviewer’s Comment:
A standardized, objective comparison of different adrenalectomy techniques should be provided, with consistent terminology and clear definitions of advantages and disadvantages.
Response:
I acknowledge the importance of presenting a structured and objective comparison of adrenalectomy techniques. To address this, I will revise the manuscript to provide a standardized comparison of SP-PRA, MP-RA, and laparoscopic adrenalectomy, ensuring consistent terminology and a balanced discussion of the advantages and limitations of each approach.
- Revising Overstated Claims
Reviewer’s Comment:
Tone down overstated claims: statements implying superiority should be revised to reflect the limitations of the available data.
Response:
I appreciate this feedback and recognize the importance of maintaining an evidence-based and balanced discussion. I will carefully review the manuscript and revise any language that implies definitive superiority of SP-PRA over other approaches, ensuring that statements reflect the current state of the literature.
Where claims are made regarding advantages (e.g., improved cosmetic outcomes, reduced postoperative pain), I will ensure that these are supported by appropriate references and framed in a way that acknowledges the need for further comparative studies.
- Balanced Discussion of Robotic Systems
Reviewer’s Comment:
The discussion of robotic systems should be more balanced, avoiding promotional language.
Response:
I agree that maintaining an objective and scientifically rigorous tone is essential. I will revise any sections where the discussion of robotic systems may appear overly promotional or subjective. Specifically, I will ensure that our descriptions of the da Vinci SP robotic system focus on its technical capabilities, clinical applications, and limitations, rather than making promotional statements.
I included a discussion of the limitations of SP robotic technology, such as instrument crowding, the learning curve, and cost considerations, to provide a more balanced perspective in discussion section.
- Explicitly Stating Patient Selection Criteria
Reviewer’s Comment:
The strict patient selection criteria for SP-PRA should be explicitly stated when making comparisons with other methods.
Response:
Thank you for your valuable feedback. The discussion on patient selection criteria is included in Chapter 4, where I address the current considerations for selecting appropriate candidates for SP-PRA. However, as SP-PRA is a relatively new surgical approach, its patient selection criteria have not yet been clearly defined. I anticipate that as surgical experience with SP-PRA increases, the indications will expand, allowing for broader application of this technique. I appreciate the reviewer’s suggestion, and I will clarify this point in the manuscript to ensure that the discussion accurately reflects the evolving nature of patient selection for SP-PRA.
- Ensuring Comparability of Outcomes Data
Reviewer’s Comment:
Ensure that claims regarding outcomes are based on truly comparable patient groups.
Response:
Thank you for your insightful feedback. I recognize the importance of comparing outcomes across similar patient populations to ensure valid conclusions. However, SP-PRA is a relatively new surgical method, and currently, there are few comparative studies evaluating its outcomes against other adrenalectomy techniques. To date, there are no randomized controlled trials (RCTs) directly comparing SP-PRA with multi-port robotic or laparoscopic adrenalectomy.
I acknowledge that comparing outcomes in truly comparable patient groups is essential to confirming the feasibility and effectiveness of SP-PRA. However, such comparisons require further research, including prospective trials and well-designed studies to control for patient selection bias and ensure meaningful comparisons.
In this review, I have focused on presenting the potential advantages and feasibility of SP-PRA based on the available literature. As the use of SP-PRA expands in clinical practice, I anticipate that more studies, including comparative analyses and RCTs, will be conducted to provide stronger evidence on its outcomes.
- Revising the Claim of "Nearly Scarless Results"
Reviewer’s Comment:
The assertion of "nearly scarless results" should be revised for accuracy.
Response:
I acknowledge this concern and will revise this statement to ensure greater accuracy and scientific precision. While SP-PRA results in a single, concealed incision, scarring can still occur, and outcomes may vary based on patient factors, incision healing, and surgical technique.
The revised statement will more accurately describe the cosmetic benefits of SP-PRA without implying an absolute absence of scarring. A possible revision may be:
Revised Statement:
" The use of a single incision, hidden within the natural contour of the back, enhances cosmetic results."
- Refining the Conclusion to Reflect Current Research Limitations
Reviewer’s Comment:
The conclusion should better reflect the current stage of research and acknowledge the need for more long-term studies before SP-PRA can be considered a standard technique.
Response:
I appreciate this suggestion and will revise the conclusion to provide a more measured assessment of SP-PRA's current status. Instead of suggesting that SP-PRA is already an established standard, I will emphasize that it is a promising technique with significant potential, but that further long-term studies are needed to validate its outcomes and expand its indications.
Revised Conclusion Excerpt:
"While SP-PRA represents an important advancement in adrenal surgery, further prospective, multicenter studies are necessary to establish its long-term oncologic outcomes, cost-effectiveness, and optimal patient selection criteria. As research continues to evolve, SP-PRA may become a preferred technique for selected patients, but additional data are needed before it can be widely considered a standard of care."
I appreciate the reviewer’s thoughtful comments, which have greatly contributed to improving the clarity and depth of our manuscript. The requested revisions will be incorporated accordingly, and I believe these enhancements will strengthen the overall impact of our work.
Sincerely,
Kwangsoon Kim MD, PhD.

Round 2
Reviewer 1 Report
Comments and Suggestions for Authors
The authors have adequately addressed my comments, and the manuscript can be accepted for publication.
Author Response
Thank you.